# Identification of Phytaspase Interactors via the Proximity-Dependent Biotin-Based Identification Approach

**DOI:** 10.3390/ijms222313123

**Published:** 2021-12-04

**Authors:** Anastasia D. Teplova, Marina V. Serebryakova, Raisa A. Galiullina, Nina V. Chichkova, Andrey B. Vartapetian

**Affiliations:** 1Faculty of Bioengineering and Bioinformatics, Lomonosov Moscow State University, Moscow 119991, Russia; anastasia_teplova@mail.ru; 2Belozersky Institute of Physico-Chemical Biology, Lomonosov Moscow State University, Moscow 119991, Russia; mserebr@mail.ru (M.V.S.); raisa-galiullina@rambler.ru (R.A.G.); chic@genebee.msu.ru (N.V.C.)

**Keywords:** plant protease, phytaspase, protein interactor, BioID, endoplasmic reticulum, calreticulin-3, senescence

## Abstract

Proteolytic enzymes are instrumental in various aspects of plant development, including senescence. This may be due not only to their digestive activity, which enables protein utilization, but also to fulfilling regulatory functions. Indeed, for the largest family of plant serine proteases, subtilisin-like proteases (subtilases), several members of which have been implicated in leaf and plant senescence, both non-specific proteolysis and regulatory protein processing have been documented. Here, we strived to identify the protein partners of phytaspase, a plant subtilase involved in stress-induced programmed cell death that possesses a characteristic aspartate-specific hydrolytic activity and unusual localization dynamics. A proximity-dependent biotin identification approach in *Nicotiana benthamiana* leaves producing phytaspase fused to a non-specific biotin ligase TurboID was employed. Although the TurboID moiety appeared to be unstable in the apoplast environment, several intracellular candidate protein interactors of phytaspase were identified. These were mainly, though not exclusively, represented by soluble residents of the endoplasmic reticulum, namely endoplasmin, BiP, and calreticulin-3. For calreticultin-3, whose gene is characterized by an enhanced expression in senescing leaves, direct interaction with phytaspase was confirmed in an in vitro binding assay using purified proteins. In addition, an apparent alteration of post-translational modification of calreticultin-3 in phytaspase-overproducing plant cells was observed.

## 1. Introduction

Subtilisin-like proteases represent the largest family of plant serine proteases that are involved in various aspects of plant life, including growth, development, senescence, programmed cell death, and stress responses [1]. Subtilases comprise over 50 members in different plant organisms, for example, 56 are found in *Arabidopsis thaliana* [2]. In relation to senescence, in *A. thaliana*, the mRNA level and proteolytic activity of subtilisin-like protease AtSBT1.4, a senescence-associated protease (SASP), was reported to increase in the leaves of senescing during the vegetative or reproductive phase [3]. Induction of subtilisin-like proteases was also observed in wheat leaves under dark- and N-starvation-induced senescence and in naturally senescing plants [4,5]. Within the subtilase family of proteolytic enzymes, phytaspases are distinguished by their strict substrate cleavage specificity after an aspartate residue, preceded by a characteristic (though degenerate) tripeptide amino acid motif [6,7,8,9]. Due to this type of recognition, similar to that of animal apoptotic proteases (caspases), phytaspases are processive (that is, they introduce single breaks into a limited number of protein substrates), rather than digestive, proteolytic enzymes. In continuation of the analogy with caspases, phytaspases were shown to be instrumental in the accomplishment of plant cell death induced by biotic and abiotic stresses [6,10,11,12,13].

Another intriguing feature of phytaspases is their dynamic localization. Phytaspase precursors are equipped with a signal peptide that guides the proenzyme to the endoplasmic reticulum with further secretion from the plant cell. In the course of this anterograde transport, phytaspase precursor becomes autocatalytically and constitutively processed, and thus activated with concomitant release and accumulation of the mature enzyme in the apoplast [6]. However, this is not the end of phytaspase transportation. Upon induction of cell death by biotic and abiotic stresses, phytaspases utilize clathrin-mediated endocytosis for retrograde transport into the plant cells [14,15].

To elucidate the molecular mechanisms of phytaspase localization dynamics, we employed an *in planta* proximity-dependent biotin identification (BioID) approach [16] to identify phytaspase interactors in *Nicotiana benthamiana* leaves. In this study, phytaspase interaction with calreticulin-3, the only plant calreticulin isoform that is up-regulated in senescing leaves [17], was uncovered in vivo and confirmed in vitro.

## 2. Results

### 2.1. Setup of the in Planta Biotinylation Assay

To obtain a *Nicotiana tabacum* phytaspase (*Nt*Phyt) derivative competent for the BioID assay, we fused the non-specific biotin ligase TurboID [18] to the *Nt*Phyt precursor. The choice for the TurboID version of biotin ligase was due to its efficient performance in plant cells relative to the previous variants of promiscuous biotin ligases [19,20,21]. As the *Nt*Phyt precursor is subjected to N-terminal processing (i.e., cleavage of the signal peptide and autocatalytic detachment of the N-terminal prodomain [6]), we fused TurboID to the carboxy terminus of the *Nt*Phyt precursor to generate *Nt*Phyt-TurboID ~120 kDa protein (Figure 1A).

A free TurboID derivative was also constructed as a control for non-specific TurboID-mediated biotinylation. To mimic *Nt*Phyt anterograde transportation, TurboID was equipped with an N-terminal signal peptide derived from *Nt*Phyt to generate SP-TurboID ~35 kDa protein (Figure 1A). In addition, both proteins were supplied with hexahistidine tags at their C-termini to allow visualization of their synthesis and accumulation in plant cells.

Synthesis of either *Nt*Phyt-TurboID or SP-TurboID in *N. benthamiana* was achieved by leaf infiltration with agrobacteria carrying a plasmid with the respective gene under the control of a strong constitutive 35 S promoter. Figure 1B demonstrates that the production of both recombinant proteins was readily detectable by Western blot analysis of extracts of infiltrated leaves.

### 2.2. Stability of NtPhyt-TurboID and SP-TurboID Proteins inside and outside the Plant Cells

To assess the stability of recombinant proteins, both within the cell and in the apoplast, apoplastic washes were obtained from leaf tissues producing each protein. The amounts of recombinant proteins in the apoplast were compared with those in the residual leaf tissue (after separation of the apoplast) by Western blotting. As shown in Figure 2A, both proteins were clearly detectable within the intracellular fraction and thus turned out to be quite stable. However, in the apoplastic fraction, only small amounts of TurboID (derived from SP-TurboID) were visible, whereas *Nt*Phyt-TurboID was undetectable.

From our previous studies, we know that the signal peptide of the *Nt*Phyt precursor guides very efficient secretion of *Nt*Phyt itself and of heterologous proteins [6,15]. Therefore, we suspected that the TurboID moiety, once secreted, is unstable in the apoplast. To verify whether *Nt*Phyt synthesized as a *Nt*Phyt-TurboID fusion protein is competent for secretion, we compared the levels of *Nt*Phyt proteolytic activity in the apoplastic and intracellular fractions using Ac-VEID-AFC, the preferred fluorogenic peptide substrate of *Nt*Phyt [6]. Figure 2B shows that the level of *Nt*Phyt activity in the apoplast of *Nt*Phyt-TurboID-producing leaves markedly exceeded the level of endogenous phytaspase activity, indicating that secretion of the enzyme was not impaired. Together with the observed stability of the *Nt*Phyt-TurboID protein within the cell, this finding is consistent with the degradation of the TurboID moiety upon secretion of the protein into the apoplast.

### 2.3. Candidate Phytaspase Interactors Revealed by BioID Approach

To identify potential *Nt*Phyt interactors, detached *N. benthamiana* leaves producing either *Nt*Phyt-TurboID or SP-TurboID were infiltrated with 200 μM biotin solution containing 500 μM ATP to support TurboID-mediated protein biotinylation and incubated for 5 h. Inspection of protein extracts obtained from these leaves using Western blot analysis with streptavidin-horseradish peroxidase (HRP) visualization revealed the presence of multiple biotinylated proteins in total leaf extracts of the control (SP-TurboID-producing) and *Nt*Phyt-TurboID-containing samples (Figure 3) and in the intracellular fractions, but not in the respective apoplastic fractions (Appendix A).

In order to simplify patterns of biotinylated proteins in the intracellular fractions, intracellular protein samples were further fractionated to obtain soluble proteins (that is, extractable in the absence of detergents) and ‘membrane’ proteins (extractable in the presence of 0.5% dodecyl maltoside detergent). After subsequent protein fractionation using ammonium sulphate precipitation and affinity chromatography using magnetic streptavidin beads, biotinylated proteins from each fraction were analysed by Western blotting with streptavidin-HRP detection. The difference between *Nt*Phyt-TurboID-containing and control samples was more clearly observed for the detergent-soluble fraction of proteins, as shown in Figure 4A. A number of protein bands that differed in the *Nt*Phyt-TurboID and control SP-TurboID samples were observed. To identify candidate protein interactors, the same protein samples were compared by gel electrophoresis and Coomassie staining. The obtained patterns of stained proteins (Figure 4B) were similar to those visualized by streptavidin-HRP detection (Figure 4A), thus confirming the successful isolation of biotinylated proteins. Proteins from bands marked with numbers in Figure 4B were subjected to trypsin hydrolysis and mass spectrometry (MS) identification.

From these analyses, we learned that band 1 (~120 kDa protein), which is present in the *Nt*Phyt-Turbo sample only, corresponds to the recombinant protein *Nt*Phyt-TurboID itself. Band 5 (~37 kDa protein) present in the control sample only corresponds to TurboID. These identifications were not surprising, as promiscuous biotin ligases are known to biotinylate themselves [18]. Band 2 (~95 kDa protein) present in the NtPhyt-TurboID sample corresponds to the endoplasmin homolog, a soluble resident of the endoplasmic reticulum. MS analysis revealed the presence of endoplasmin in the same position of the control (SP-TurboID) lane as well, although this was not evident from the protein patterns in Figure 4B. Bands 3 (~56 kDa protein) and 4 (~40 kDa protein) observed predominantly in the phytaspase-expressing sample were identified as ribulose-1,5-bisphosphate carboxylase/oxygenase large subunit and an uncharacterized protein LOC109229657, respectively. Analysis of the amino acid sequence of this latter protein revealed the presence of a signal peptide and a saposin B-type domain in its structure. Saposin B-type domains in known animal and plant membrane-interacting proteins have been reported to destabilize lipid membranes, with or without membrane permeabilization [22].

### 2.4. Prolonged Incubation of Leaves with Biotin Revealed Additional Potential Phytaspase Interactors

To increase the sensitivity of the assay, *N. benthamiana* leaves producing either *Nt*Phyt-TurboID or SP-TurboID proteins were incubated with biotin for 16 h. Fractionation of leaf extracts was performed, as described in the previous section. Representative patterns of biotinylated proteins from the detergent-soluble fraction isolated by affinity chromatography from both samples and stained with Coomassie are shown in Figure 5. Apart from MS re-identification of *Nt*Phyt-TurboID Figure 5, band 1) and the endoplasmin homolog (Figure 5, band 2), two novel candidate interactors were identified. Band 3 in the *Nt*Phyt-TurboID sample corresponds to BiP (immunoglobulin binding protein), an endoplasmic reticulum luminal binding protein belonging to the Hsp70 protein family. Although a faint band with similar electrophoretic mobility was observed in the control sample (band 3c), MS analysis failed to identify BiP as a TurboID interactor. Thus, BiP appears to specifically interact with *Nt*Phyt.

Another candidate protein (band 4 in Figure 5, ~52kDa) was identified as calreticulin-3, an endoplasmic reticulum lumen protein [23] known to function as a chaperone and a calcium-binding protein. Calreticulin-3 was identified in both the *Nt*Phyt-TurboID and control samples (bands 4 and 4c, respectively), although its electrophoretic mobility appeared to be slightly slower in the control. This might be indicative of alterations in post-translational modification in phytaspase-overproducing leaves.

Table 1 summarizes our findings in identifying possible *Nt*Phyt interactors using the BioID approach. The majority of proteins identified (endoplasmin, BiP, and calreticulin-3) represent known residents of the endoplasmic reticulum lumen that function as chaperones [23]. Of these, only BiP appears to represent a “true” candidate phytaspase interactor, as it was not detected in the control (SP-TurboID-producing) sample. However, for calreticulin-3, the impact of phytaspase may also be expected, as this protein displays slightly altered mobility in the *Nt*Phyt-TurboID-producing samples. Another example of a specific interactor absent in control probes is an uncharacterized saposin B-type domain-containing protein. As for the Rubisco subunit, a protein of chloroplast origin, we do not yet know whether its identification was due to the enormous abundance of this protein in plant cells, as the presence of the corresponding biotinylated protein was not evident in Figure 4a.

For proteins listed in Table 1, sequence coverage with the MS-identified peptides is given in Appendix A.

### 2.5. Calreticulin-3 Interacts with NtPhytaspase In Vitro

To verify the results of the in planta BioID approach and to clarify whether calreticulin-3 might represent a true phytaspase partner, the possibility of direct interaction between purified *N. tabacum* calreticulin-3 and *Nt*Phyt was assessed. Recombinant *N. tabacum* calreticulin-3, in which the signal peptide was substituted with a His6 tag (His-CRT3), was overproduced in *E. coli* cells and immobilized on Ni-NTA agarose. Untagged *Nt*Phyt was overproduced and affinity-purified using a specific and reversible *Nt*Phyt inhibitor from *N. benthamiana* leaves [6]. Upon incubation with immobilized calreticulin-3, the amount of *Nt*Phyt proteolytic activity in the flow-through (unbound enzyme) and in the eluate from the column with EDTA-containing buffer (releasing the potential His-CRT3–*Nt*Phyt complex) was quantitatively determined. Ni-NTA agarose resin preincubated with an equivalent amount of lysate of vector-only transformed *E. coli* cells served as a control. As shown in Figure 6A, phytaspase activity was reduced in the flow-through from the calreticulin-3-containing resin, relative to the control sample, while it was enhanced in the eluate (Figure 6B), thus providing evidence for direct calreticulin-3-phytaspase interaction.

## 3. Discussion

BioID is a powerful tool for the identification of protein interactors in living cells by means of proximity-dependent biotinylation, which was recently successfully applied in plant systems [19,20,21,24,25]. We used this approach to identify candidate protein partners of *N. tabacum* phytaspase, a subtilisin-like protease involved in the accomplishment of plant cell death induced by biotic and abiotic stresses [6,10,12]. As with the majority of plant subtilases, phytaspase is secreted into the apoplast of healthy plant tissues. However, a peculiarity of phytaspase is its retrograde transport from the apoplast back into plant cells that employs clathrin-mediated endocytosis and occurs upon the induction of programmed cell death [14,15]. By utilizing the BioID approach, we were hoping to identify proteins involved in phytaspase mobility.

In this study, we used a TurboID version of biotin ligase, as previous studies have shown that TurboID outperforms other non-specific biotin ligases for studies in plant cells [19,20,21]. For our search of *Nt*Phyt interactors, a closely related *N. benthamiana* species was chosen as a host, as it is a widely used model plant, for which the BioID approach has been optimized [20]. However, we discovered a limitation of our scheme associated with an apparent instability of the TurboID moiety within the aggressive *N. benthamiana* apoplast environment. This might explain why we did not detect the extracellular partners of phytaspase. On the other hand, the absence or low abundance of stable phytaspase-interacting proteins in the apoplast cannot be excluded.

Intracellular *Nt*Phyt-TurboID was fairly stable, and this allowed us to identify a set of candidate protein interactors within the plant cell. Interestingly, three of these proteins (endoplasmin, BiP, and calreticulin-3) represent soluble residents of the lumen of the endoplasmic reticulum. They are known to perform chaperone and protein quality control functions for newly synthesized proteins targeted to the endoplasmic reticulum and serve as calcium ion stores (calreticulin) [23]. As the phytaspase precursor protein has to pass through the endoplasmic reticulum on the route of the enzyme from the plant cell, phytaspase contacts with the protein quality control system are intuitively understandable. However, it is not uncommon, both in plants and in animals, to find these endoplasmic reticulum “residents” at various additional locations, such as the plasma membrane, cell surface, nucleus, and cytoplasm, where these proteins were reported to perform important tasks [26,27,28,29]. Therefore, the possibility of phytaspase interaction with these proteins at locations other than the endoplasmic reticulum cannot be completely dismissed.

The specificity of phytaspase interaction with newly detected candidate proteins also deserves comment. Given that phytaspase is synthesized as an *Nt*Phyt-TurboID fusion protein, who is involved in the interaction with the partner: phytaspase, TurboID, or both? Employment of the described SP-TurboID control construct was helpful in cases where the protein of interest was absent in the control track, such as BiP and saposin B-type domain-containing protein. However, in the case of endoplasmin and calreticulin-3, where the corresponding protein was identified in the control sample as well, the situation was less clear. The hint that phytaspase may be involved in the interaction with calreticulin-3 followed indirectly from a slightly altered electrophoretic mobility of the partner (possibly due to some post-translational modification) in the case of *Nt*Phyt-TurboID production. Therefore, calreticulin-3 was chosen as a target to verify the conclusions obtained using the BioID approach and to address the issue of specificity in this complicated case. Our in vitro assay with the purified *Nt*calreticulin-3 and *Nt*Phyt proteins supported the conclusion that these two proteins are capable of direct interaction. Calreticulin-3 is a member of calreticulin protein family consisting in plants of 3 members which have acquired specialized functions [17,30]. Plant calreticulin-3 has been reported to play important roles in maintaining plant innate immunity and in protecting plants against fungal and bacterial pathogens [31,32,33,34,35]. Of note, an enhanced expression of *Arabidopsis thaliana CRT-3* gene in senescing leaves was reported, that contrasted markedly the expression patterns of the two other family members, which were much reduced [17]. As our BioID analysis identified calreticulin-3, but not the other family members, as a phytaspase partner, a functional role of calreticulin-3–phytaspase interaction in plant innate immunity and senescence can be expected. An intriguing question still remains as to what kind of alteration in post-translational modification occurs to calreticulin-3 upon phytaspase overproduction in plant cells.

Yet, another potential partner of phytaspase identified in this study, saposin B-type domain-containing protein, also appears to be an interesting candidate. Proteins that harbour such a domain have been reported to localize to the endoplasmic reticulum and plasma membrane. The saposin B domain is capable of associating with membrane lipids and causes distortion and, in some cases, membrane permeabilization [22]. We hypothesize that the interaction with saposin B-containing protein might be related to retrograde trafficking of phytaspase in stressed plant cells, a possibility that is worth addressing in the future.

## 4. Materials and Methods

### 4.1. Plant Growth Conditions

*N. benthamiana* plants were grown at 25 °C in soil in a controlled environment under a 16/8 h day/night cycle. For transient protein production, *Agrobacterium tumefaciens* GV3101 cells transformed with the respective plasmid (see below) were mixed with an equal amount of agrobacteria bearing the p19 suppressor of silencing and infiltrated into leaves of six-week-old plants using a blunt syringe.

### 4.2. Plasmid Construction

The ORF of TurboID was amplified by polymerase chain reaction (PCR) using the R4pGWB601_UBQ10p-Turbo-NES-YFP plasmid (a gift from D. Bergmann; Addgene plasmid #127366) using primers TurboID_Bam_dir and TurboID_His6_z_Sac_rev (Appendix A). Six His codons were included in the reverse PCR primer to permit the detection of the recombinant protein. The PCR product (c. 1000 bp long) was cloned between the BamHI and SacI sites of *Nt*Phyt cDNA-carrying pLEX7000 expression plasmid [36] modified to eliminate an additional BamHI site in the polylinker. The resultant pLH-*Nt*Phyt-Turbo-His construct contained the *Nt*Phyt-TurboID ORF (the two moieties are separated with a Gly-Gly-Ser spacer) under the control of the CaMV 35S promoter. Insertion of the same PCR product between the BamHI and SacI sites of the pLH_SP_mRFP_His plasmid [15] produced the pLH_SP_Turbo_His construct expressing the SP-TurboID ORF under the control of the 35S promoter.

The ORF of *N. tabacum* calreticulin-3 (c. 1300 bp) was amplified by PCR on total *N. tabacum* cDNA using Phusion Hot Start II High-Fidelity DNA Polymerase (Thermo Scientific, Waltham, MA USA) and CRT_Kpn_Nco_dir and CRT_Bam_rev primers (Appendix A), cloned into the pSL1180 vector between the KpnI and BamHI sites, and sequenced to confirm its identity. To construct a plasmid for bacterial production of calreticulin-3 with the signal peptide substituted with the His tag, the calreticulin-3 cDNA fragment was PCR-amplified using CRT_LF_Apa_Nde_dir and CRT_Bam_rev primers. The 1200 bp PCR product was then inserted downstream of and in frame with six His codons between the NdeI and BamHI sites of the pET28a (+) vector (Novagen) to generate the pET_His_LF Calreticulin plasmid encoding His-CRT fusion protein.

### 4.3. In Planta Proximity-Dependent Biotinylation Assay

Leaf discs from *N. benthamiana* plants transiently producing either *Nt*Phyt-Turbo or SP-Turbo protein were prepared 2–3 d post agroinfiltration. Discs were then vacuum infiltrated with 200 μM biotin (Sigma) solution in water containing 500 μM ATP and 1.25 mM magnesium acetate, and incubated at room temperature for the time periods indicated in the figure legends.

### 4.4. Fractionation of Biotinylated Proteins

After incubation with biotin, initial steps of leaf protein isolation from different subcellular compartments were performed as described previously [15]. Briefly, apoplastic washes were obtained by low-speed (2000 *g*) centrifugation of *Nt*Phyt-TurboID- and SP-TurboID-producing leaf discs (typically, 5 *g* of leaf material) at 4 °C for 10 min and diluted with the ICL buffer (10 mM Tris-HCl, pH 8.8, 0.2 M NaCl, 30 mM magnesium chloride (MgCl_2_), 0.2 M sucrose, and 10 mM 2-mercaptoethanol), containing protease inhibitors aprotinin (2 µg/mL), leupeptin (6 µg/mL), chymostatin (6 µg/mL), E64 (6 µg/mL), 4-(2-aminoethyl)benzenesulfonyl fluoride (25 µg/mL), and 2 mM EDTA. The residual leaf material was frozen in liquid nitrogen and disrupted in a Minilys homogenizer (Bertin Instruments, Montigny-le-Bretonneux, France) using 1.4 mm ceramic beads with two 20 s bursts. An additional 20 s burst was performed after suspending the sample in 15 mL of ICL buffer, and the sample was incubated on ice for 15–30 min. Water-insoluble material was pelleted by centrifugation for 10 min at 10,000 *g* at 4 °C, and the supernatant obtained represented the water-soluble fraction of the intracellular proteins.

Pellets obtained in the previous step contained intracellular proteins that resisted extraction with an aqueous buffer. For their isolation, precipitates were re-suspended in 15 mL of ICL buffer supplemented with 0.5% n-dodecyl-β-D-maltopyranoside (dodecyl maltoside, Anatrace) and protease inhibitors, incubated for 10 min on ice, and the detergent-solubilized protein fraction was obtained as a supernatant after centrifugation for 10 min at 10,000 *g* at 4 °C.

Both water-soluble and detergent-soluble proteins were further fractionated by ammonium sulphate precipitation. The protein fractions that precipitated within the 0–30, 30–50, and 50–70% intervals of (NH_4_)_2_SO_4_ saturation were dissolved in 1.5 mL of 50 mM Tris-HCl buffer (pH 7.5) containing protease inhibitors. Biotinylated proteins from these samples were then separated by affinity chromatography using Streptavidin magnetic polymer resin (UBPBio), according to the manufacturer’s protocol with modifications. Prior to separation, protein samples were supplied with 1% sodium dodecyl sulphate (SDS), boiled for 5 min, cooled on ice, and diluted with an equal volume of 50 mM Tris-HCl buffer (pH 7.5) containing 0.05% Tween 20 and 1 M NaCl. Magnetic beads (200 µL) were washed with 2 × 1 mL of 50 mM Tris-HCl buffer (pH 7.5) containing 0.05% Tween 20 and 0.5 M NaCl prior to their addition to protein samples and incubated for 4 h at room temperature with rotation. Beads with bound biotinylated proteins were then separated with a magnet, washed with 3 × 1 mL of 50 mM Tris-HCl buffer (pH 7.5) containing 4 M urea, 0.5 M NaCl, 1 mM dithiothreitol, 0.05% Tween 20, and biotinylated proteins were eluted from the beads with 400 μL of elution mix (2% SDS, 6 M urea, and 10 mM dithiothreitol) by incubating the samples at 98 °C for 5 min.

The eluted protein samples were concentrated by overnight precipitation with 80% acetone. Precipitated proteins were dissolved by pipetting in 50 mM Tris-HCl buffer (pH 8.0) containing 1% SDS and alkylated by incubation with 10 mM iodoacetamide for 30 min at room temperature. The reaction was stopped by boiling with the sample buffer, and the protein samples were fractionated by SDS gel electrophoresis. Separated proteins were electrophoretically transferred onto polyvinylidene fluoride (PVDF) membranes and incubated with streptavidin-horseradish peroxidase conjugate (GE Healthcare, for biotinylated proteins) or with HisProbe reagent (Thermo Scientific, for His-tagged proteins). Chemiluminescence detection was performed with ECL Western Lightning Plus reagent (PerkinElmer, Waltham, MA, USA) using the ChemiDoc Imaging System (Bio-Rad Laboratories, Hercules, CA, USA). Data were reproducible over three independent experiments.

### 4.5. Activity Assay of NtPhyt

The proteolytic activity of phytaspase in apoplastic washes, intracellular protein fractions (equivalent to 3 mg of leaf material), and in sub-fractions of the calreticulin-3–phytaspase binding assay (1.5 ng *Nt*Phyt in total) was determined using Ac-VEID-AFC [AFC, 7-amino-4-(trifluoromethyl) coumarin] fluorogenic peptide substrate (California Peptide), as described in [6]. To perform kinetic measurements of relative fluorescence increase, protein samples were 5-fold diluted with B1 buffer (20 mM MES, 2 mM dithiothreitol, 0.1% Tween 20, and 5% glycerol), pH 5.5, containing 0.5 M NaCl at 28 °C before activity measurements. The peptide substrate was used at a final concentration of 20 µM. A FLUOstar OPTIMA reader (BMG Labtech, Ortenberg, Germany) equipped with 405 nm excitation and 520 nm emission filters was used to quantitate fluorescence intensities. Data are presented as the means of three independent experiments.

### 4.6. MS Analyses

Matrix-assisted laser desorption ionization-time of flight (MALDI-TOF) MS analysis of tryptic digests was performed on an UltrafleXtreme MALDI-TOF/TOF mass spectrometer (Bruker Daltonics, Bremen, Germany) equipped with an Nd laser by detection of MH+ molecular ions. Pieces of about 2 mm^3^ of protein-containing gel were de-stained twice with 40% aqueous acetonitrile solution, dehydrated with 100 mL of 100% acetonitrile, and rehydrated with 5 mL of the digestion solution (pH 6) containing 15 µg/mL sequencing grade trypsin. Digestion was performed at 37 °C for 4 h. Proteolysis was stopped by the addition of 10 µL of 0.5% trifluoroacetic acid (TFA). An aliquot of 0.5 μL was mixed with 1 μL of 2,5-dihydroxybenzoic acid solution (40 mg/mL in 30% acetonitrile, 0.5% trifluoroacetic acid (TFA)). The spectra of tryptic digests were recorded in reflector mode, and the accuracy of the monoisotopic mass peak measurement was within 30 ppm. Mass spectra were processed using FlexAnalysis 3.3 software (Bruker Daltonics, Bremen, Germany). Proteins were identified using the home database, which was preloaded with *N. benthamiana* NbDE dataset [37] and Mascot combined peptide mass fingerprint + MS/MS search program (Mascot version 2.3.02). The search allowed for one possible missed cleavage, Met oxidation, and Cys-carbamidomethylation. Protein scores greater than 64 were considered to be significant (*p* < 0.05).

### 4.7. Calreticulin-3–Nt Phyt In Vitro Binding Assay

*E. coli* BL21 (DE3) cells transformed with pET28a-based construct encoding His-CRT3 fusion protein were grown in 20 mL of LB medium containing kanamycin (25 μg/mL) at 37 °C with shaking. Exponentially growing cells were shifted to 18 °C, induced with 1 mM isopropyl-β-D-thiogalactoside, and further incubated overnight at 18 °C with shaking. Cells (approx. 75 mg) were pelleted by centrifugation (2000 × g for 15 min at 4 °C), resuspended in 5 mL of sonication buffer (50 mM sodium phosphate, 300 mM NaCl, pH 8.0) containing protease inhibitors aprotinin (2 μg/mL), leupeptin (25 μg/mL), chymostatin (100 μg/mL), and phenylmethylsulphonyl fluoride (10 μM), sonicated on ice by five 20 s bursts, and centrifuged at 10,000 *g* for 20 min at 4 °C. The cleared lysate was incubated with Ni-NTA agarose (Qiagen; 200 μL of an aqueous suspension per experiment) pre-washed with sonication buffer for 1 h at 4 °C with rotation. The resin was then washed with 3 × 1 mL of Wash8.0 buffer (50 mM sodium phosphate, pH 8.0, 1 M NaCl, 0.05% Tween 20, 10% glycerol), then with 5 × 0.5 mL of Wash6.4 buffer (50 mM sodium phosphate, pH 6.4, 300 mM NaCl, 0.05% Tween 20, 10% glycerol), and stored on ice. Ni-NTA agarose sample treated in an analogous fashion with a lysate of *E. coli* BL21 (DE3) cells transformed with an empty pET28a (+) vector served as a control in the *Nt*Phyt binding experiments.

Untagged *Nt*Phyt was affinity-purified from the apoplast of *Nt*Phyt-overproducing *N. benthamiana* leaves using the reversible biotinylated peptide aldehyde inhibitor Bio-TATD-CHO (Bachem) as described in [6]. To assess CRT3-*Nt*Phyt binding, an aliquot (5 μL) of Ni-NTA agarose with immobilized His-CRT3 (approx. 0.5 μg protein per binding experiment), or an equivalent amount of the control resin, was washed with 4 × 50 μL of 20 mM MES buffer (pH 6.5) and incubated with *Nt*Phyt (15 ng) in a total volume of 30 μL of the same buffer for 3 h on ice. After centrifugation at 12,000 *g* for 15 s, unbound *Nt*Phyt contained in the flow-through and in 2 × 30 μL washes with 20 mM MES buffer (pH 6.5) was combined, and the resin-bound proteins were eluted by 10 min incubations on ice with 3 × 30 μL of B1 buffer, pH 5.5, containing 50 mM NaCl and 100 mM EDTA. Aliquots (10%) of the combined eluates and unbound proteins were used to determine *Nt*Phyt activity. Aliquots of eluates were analysed by SDS-gel electrophoresis with Coomassie staining to evaluate the amount of His-CRT3 in the samples. Experiments with *Nt*Phyt binding were performed in triplicate. Data are presented as means from these experiments. Statistical analysis was performed using GraphPad Prism 8.0.1 software (GraphPad Software, San Diego, CA, USA). Statistical significance was analysed using the unpaired t-test. The level of significance is indicated by asterisks in Figure 6.

## Figures and Tables

**Figure 1 ijms-22-13123-f001:**
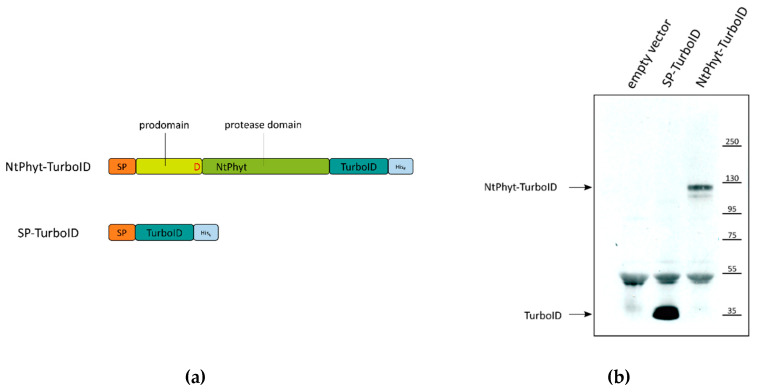
*Nicotiana tabacum* phytaspase (*Nt*Phyt)-TurboID and signal peptide (SP)-TurboID proteins utilized for the identification of phytaspase interactors. (**a**) Schematic representation of recombinant proteins. Non-specific biotin ligase TurboID fused to SP derived from *Nt*Phyt served as a control for *in planta* biotinylation accomplished by *Nt*Phyt-TurboID protein. Both proteins bear His tags to permit their detection using Western blotting analysis. (**b**) Transient production of *Nt*Phyt-TurboID and SP-TurboID in *N. benthamiana* leaves. Protein extracts from *Nt*Phyt-TurboID-producing, SP-TurboID-producing and control (vector) leaves were fractionated by sodium dodecyl sulphate (SDS) gel-electrophoresis and analysed by Western blotting with HisProbe detection. Positions of molecular weight protein markers are indicated on the right.

**Figure 2 ijms-22-13123-f002:**
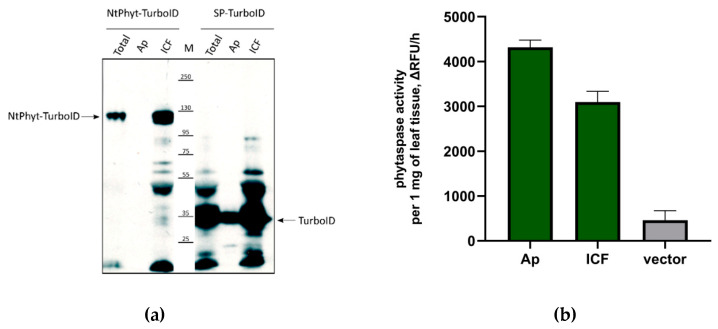
Evaluation of the stability of *Nt*Phyt-TurboID and SP-TurboID proteins in the apoplast and inside the plant cells. (**a**) For *Nt*Phyt-TurboID- and SP-TurboID-producing leaves, proteins in the apoplastic washes (Ap) and intracellular fractions (ICFs) were analysed by Western blotting with HisProbe detection. ‘Total’ represents leaf extracts without fractionation. Equal amounts of leaf tissues (5 mg) were taken for protein analyses, and equivalent 15 μL aliquots of the subcellular fractions were loaded on the gel. M, molecular weights of protein markers. Arrows indicate positions of the recombinant proteins. (**b**) Measurement of phytaspase proteolytic activity in the extracellular (Ap) and intracellular (ICF) fractions obtained from *Nt*Phyt-TurboID-producing leaves. ‘Vector’ total protein sample from leaves infiltrated with agrobacteria carrying the empty vector. Ac-VEID-AFC (20 μM) was used as the phytaspase substrate for quantitative assessment of phytaspase proteolytic activity. Relative rates of hydrolysis were determined as an increase of relative fluorescence units per hour (deltaRFU/h). Enzymatic activities were normalized by the weight of leaf tissues taken for analysis. Data represent the mean of three independent experiments ± standard deviation (SD).

**Figure 3 ijms-22-13123-f003:**
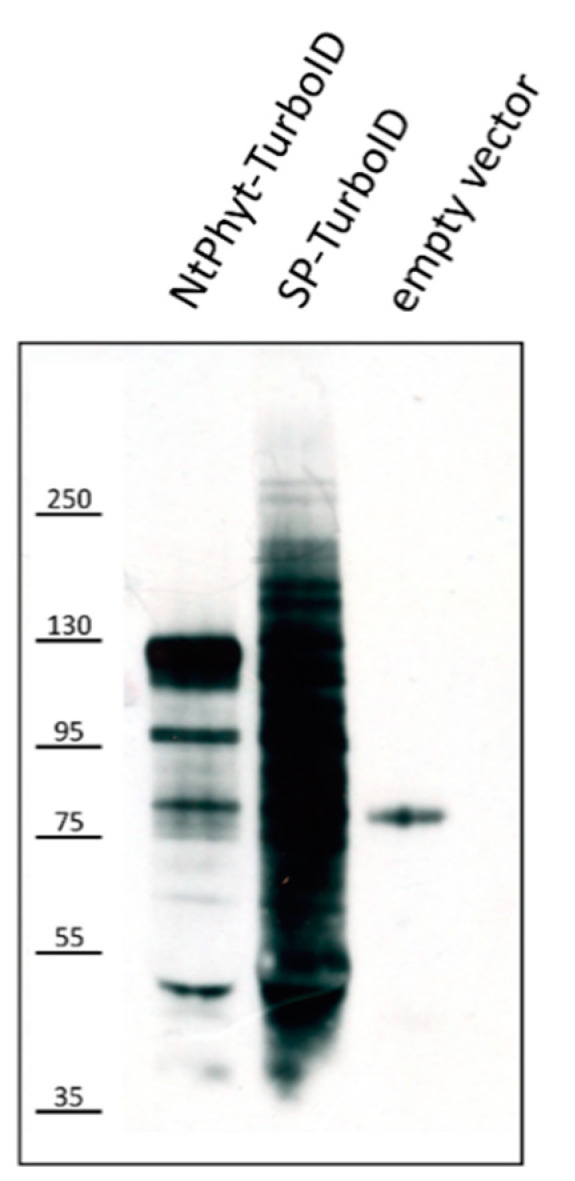
Protein biotinylation in *N. benthamiana* leaves producing either *Nt*Phyt-TurboID or SP-TurboID proteins. Western blotting analysis with streptavidin-horseradish peroxidase (HRP) detection of total proteins biotinylated in *N. benthamiana* leaves producing either *Nt*Phyt-TurboID or SP-TurboID proteins. ‘Vector,’ total protein sample from leaves infiltrated with agrobacteria carrying the empty vector. Equal amounts of leaf tissues (5 mg) were taken for analysis, and equivalent 15 μL aliquots of the subcellular fractions were loaded. Positions of molecular weight protein markers are indicated on the left.

**Figure 4 ijms-22-13123-f004:**
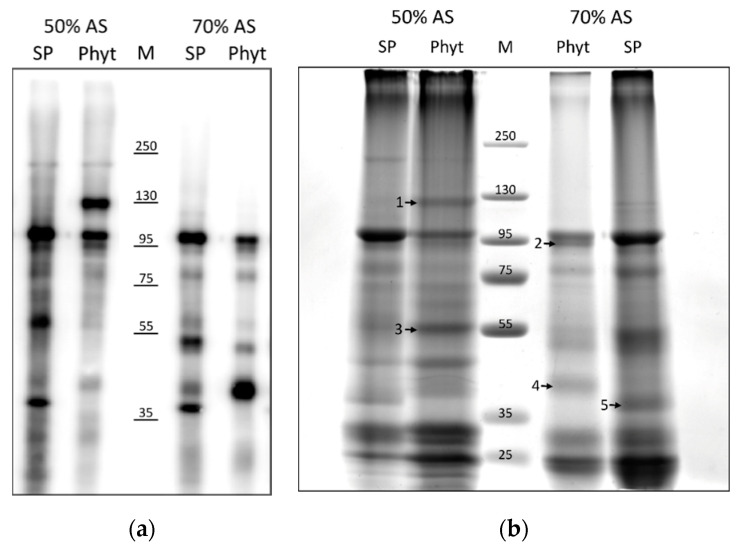
Visualizing the candidate protein interactors of *Nt*Phyt. Proteins biotinylated in *Nt*Phyt-TurboID- or SP-TurboID-producing leaves during the incubation of leaf samples with 200 μM biotin for 5 h (Phyt and SP, respectively) were isolated from the respective intracellular fractions (obtained from equal 5 g amounts of leaf tissues) by extraction in the presence of 0.5% dodecyl maltoside. Upon ammonium sulphate (AS) fractionation, biotinylated proteins precipitated with 50% AS and 70% AS were separated by affinity chromatography on streptavidin magnetic beads and analysed by SDS gel electrophoresis. (**a**) On-blot detection of biotinylated proteins using streptavidin-HRP. One % of the affinity purified protein samples in 4 μL aliquots were loaded. (**b**) The rest of the samples, after acetone precipitation, were dissolved in 45μL of SDS-containing buffer and analysed by SDS gel electrophoresis. Coomassie blue staining of the 6–16% gradient polyacrylamide gel. Arrows with numbers indicate the bands chosen for protein identification by mass spectrometry (MS) analysis. M, molecular weights of the protein markers.

**Figure 5 ijms-22-13123-f005:**
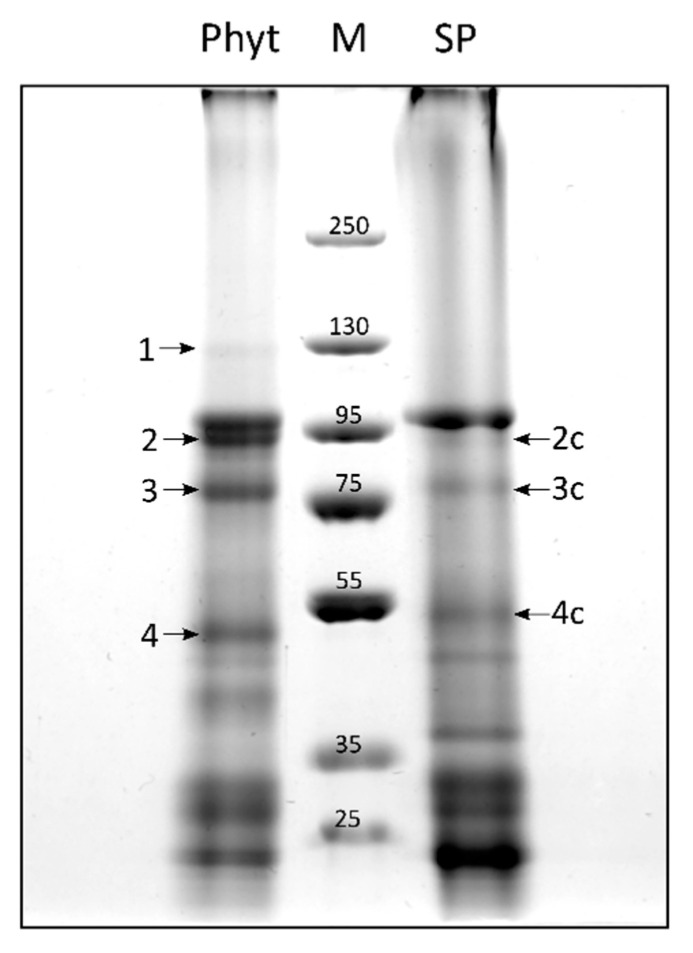
Longer incubation of leaf samples with biotin reveals new candidate phytaspase interactors. After prolonged (16 h) incubation of leaf samples with 200 μM biotin, proteins were extracted from the intracellular fractions of *Nt*Phyt-TurboID- or SP-TurboID-producing leaves (Phyt and SP, respectively; obtained from equal 5 g amounts of leaf material) in the presence of 0.5% dodecyl maltoside. Biotinylated proteins that precipitated within the 50–70% interval of ammonium sulphate saturation were further affinity-purified using streptavidin magnetic beads. The eluted protein samples were concentrated by acetone precipitation, dissolved in 45 μL of SDS-containing buffer and analysed by SDS electrophoresis in a 6–16% gradient polyacrylamide gel. Proteins were visualized after Coomassie Blue staining. M, molecular weights of the protein markers. Arrows with numbers indicate the bands chosen for subsequent protein identification.

**Figure 6 ijms-22-13123-f006:**
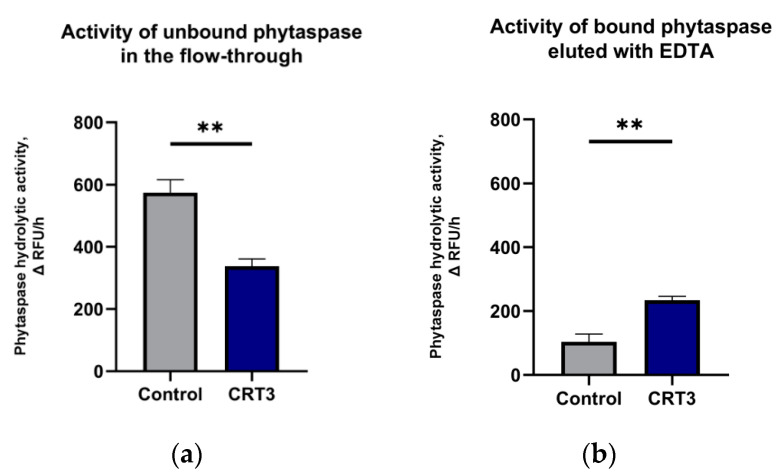
Evidence for direct in vitro interaction of *Nt*Phyt with calreticulin-3. Recombinant *N. tabacum* calreticulin-3 bearing an N terminal His6 tag (His-CRT3) was produced in *Escherichia coli* cells and immobilized on Ni-nitrilotriacetic acid (Ni-NTA) agarose (CRT3, approx. 0.5 μg protein per binding experiment). Ni-NTA agarose resin preincubated with an equivalent amount of lysed vector-only transformed *E. coli* cells was used as a control (control). Upon incubation of each resin with *Nt*Phyt for 1 h, *Nt*Phyt proteolytic activity in the flow-through fractions (**a**) and in the eluates from the column with EDTA-containing buffer (**b**) was fluorometrically determined using 20 μM Ac-VEID-AFC as a phytaspase substrate. Data represent the mean ± SD of three independent experiments. Significant differences to the control are shown as ** *p* < 0.01. The two-tailed *p* value is 0.0010 for (**a**) and 0.0011 for (**b**) (unpaired *t*-test).

**Table 1 ijms-22-13123-t001:** Biotinylated proteins–candidate phytaspase interactors identified in this study.

Name.	Identification	Mr	Fraction	Presence in the Control
Endoplasmin	XP_019239585.1	95 kDa	detergent soluble	Yes
BiP	XP_016484998.1	78 kDa	detergent soluble	No
			water soluble	
Rubisco (large subunit)	NP_054507.1	56 kDa	detergent soluble	No
Calreticulin-3	XP_016452363.1	52 kDa	detergent soluble	Yes, lower mobility in the control
Saposin B type domain containing protein	LOC109229657 XP_019250726.1	40 kDa	detergent soluble	No

## Data Availability

Exclude.

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
