# Peer review of "Identification of Phytaspase Interactors via the Proximity-Dependent Biotin-Based Identification Approach"

_ijms, 2021, doi:10.3390/ijms222313123_

Round 1
Reviewer 1 Report
The paper by Teplova et all. aimed at identification of proteins interacting with phytaspase. To do this, the Authors used a very interesting technique basing on biotynylation of binding partners. This approach led to selection of several cellular proteins which potentially may affect phytaspase trafficking.
The results are potentially valuable, but the way they were presented does not allow for drawing clear conclusions.
First of all, it appears that the amounts of protein in the analyzed samples were not normalized. Each sample should contain equal amount of protein or should represent protein from equal amount of plant tissue. Otherwise, quantitative comparisons of enzymatic activity or finding differences between samples are not possible.
Fig 2a, 3-5 – Please, provide information about protein content in each line. Were equal amounts of protein resolved in each line (or were they derived from the same amount of plant tissue?)
Fig 2b – Please recalculate the phytaspase activity per amount of protein in each sample (or amount of plant tissue).
In Fig 4b – a protein band no3 is not visible in part 4a. Does it mean it was not biotynylated? If so, is it a partner for phytaspase?
In the Materials and Methods section some descriptions lack important details which are necessary to repeat experiments:
method 4.4 – the amounts of plant material and volumes were not provided. What was the final protein concentration in each extract and after affinity chromatography? How many times the experiment was repeated?
Method 4.5 – how much protein was used in each experiment?
Method 4.6 – How much protein was loaded to NiNTA resin? How much resin was used?
Method 4.7 – what was the final OD of bacterial culture (or weight of cells after centrifugation)?
Line 447 ...an aliquot of NiNTA agarose...” please, provide the volume; how much Nphyt was loaded?
Please, check also the text to correct the typo errors.
Author Response
We would like to thank the reviewers for their time and helpful comments. Below we address each of the comments one by one.
Reviewer 1 Comments for the Author
The results are potentially valuable, but the way they were presented does not allow for drawing clear conclusions.
First of all, it appears that the amounts of protein in the analyzed samples were not normalized. Each sample should contain equal amount of protein or should represent protein from equal amount of plant tissue. Otherwise, quantitative comparisons of enzymatic activity or finding differences between samples are not possible.
Fig 2a, 3-5 – Please, provide information about protein content in each line. Were equal amounts of protein resolved in each line (or were they derived from the same amount of plant tissue?)
Answer: We agree with the reviewer that it is essential to compare equal aliquots of the samples. Actually, in all our experiments, though the exact protein content was not determined directly, equal amounts of leaf tissues were taken for protein and enzymatic activity analyses. We now added the appropriate statements to legends to Figures 2, 3, 4, 5 in the revised version of the manuscript.
Some differences in band intensities between the ‘Total’ and ‘ICF’ samples in Figure 2a could thus be due to uneven protein extractability from the total and sub-fractionated samples.
Also of note, although equal amounts of leaf tissues were taken for analyses, there is a clear difference in signal intensities between the NtPhyt-TurboID and SP-TurboID samples in Figure 3. This was due to the more efficient in planta production of free TurboID, relative to the NtPhyt-TurboID fusion protein, as is evident from Figures 1b and 2a.
Fig 2b – Please recalculate the phytaspase activity per amount of protein in each sample (or amount of plant tissue).
Answer: Done; please see legend to Figure 2b in the revised version of the manuscript. Figure 2b itself does not need recalculation as the data were initially normalized to sample weights – please see our response to the previous question.
In Fig 4b – a protein band no3 is not visible in part 4a. Does it mean it was not biotynylated? If so, is it a partner for phytaspase?
Answer: We agree with this comment of the reviewer, and this was the reason for us to state that “we do not yet know whether its identification was due to the enormous abundance of this protein (Rubisco)” (lines 189-190 in our original text). Yet we felt that we cannot simply ignore mass spectrometric identification of this protein. We now added the argument put by the reviewer; please see p. 8 in the revised version of the manuscript.
method 4.4 – the amounts of plant material and volumes were not provided. What was the final protein concentration in each extract and after affinity chromatography? How many times the experiment was repeated?
Answer: The available data were added to section 4.4. As noted above, the exact protein concentrations were not determined after each fractionation step. Yet equal amounts of starting leaf material and equal sub-fraction aliquots were taken for analysis. Data were reproducible over three independent experiments. Please see p. 11, 12 in the revised version of the manuscript.
Method 4.5 – how much protein was used in each experiment?
Answer: Done; please see p. 12 in the revised version of the manuscript.
Method 4.6 – How much protein was loaded to NiNTA resin? How much resin was used?
Answer: Done; please see p. 13 in the revised version of the manuscript.
Method 4.7 – what was the final OD of bacterial culture (or weight of cells after centrifugation)?
Answer: Done; please see p. 13 in the revised version of the manuscript.
Line 447 ...an aliquot of NiNTA agarose...” please, provide the volume; how much Nphyt was loaded?
Answer: Done; please see p. 13 in the revised version of the manuscript.
Please, check also the text to correct the typo errors.
Answer: Done.
Reviewer 2 Report
This appears as a well made, easy to follow and understand work, that provides new and interesting insights on the protein partners of the subtilisin-like phytaspase, a protease involved in programmed cell death in plants. The works seems to be correctly made and contributes to the advancement of its research field. There are a series of points of this work on which the present reviewer considers that should be revised for its improvement and a better understanding:
1.-At the Introduction, lines 35-38, although it could be deduced from the context that SASP, the senescence-associated protease, mentioned in these lines is a subtilisin-like protease, it would be desirable to explicitly indicate such identification there, at such point.
Also, at lines 40-44 it is commented that phytaspases are distinguished by their strict cleavage specificity after an aspartate residue, after a tripeptide, motif; here, again, it would be wise to add the word "substrate" before "cleavage specificity" (or equivalent) to avoid possible doubts on the receiver of such action, and more if it is taken into account that such enzymes experiments auto-processing events.
Also, in the same paragraph, at lines 43-44, it is commented that it can be said that such proteases, phytaspases, "are processive rather than digestive" due to such type of recognition. Given that "processivity" is a term that, in such context, could have several variants and meaning, it would be advisable to better explain it there and/or in another point of the manuscript. Also, the term "makes" there is a bit confusing, and it would be better substitute it.
2.-At different points of the work there are citations and comments on subtilases/phytaspases from A. thaliana and from N. tabacum. However, the research work is actually performed on N. benthamiana leaves. Authors should explain at the Introduction (and /or perhaps also at the Discussion) why they finally selected N. benthamiana to do the work
3.-Figure1 displays the sketch of the proteins and protein constructs used in this work for the identification of phytaspase interactors. However, the detailed structure of such constructs, grossly exemplified by the depiction of "TurboID" domain in the sketches of N-PhyTurboID and SP-turboID, in subFig 1a, does not provide enough detailed information about, requiring the need to get such information elsewhere. Given that such Figure is not crowded, this reviewer would suggest the inclusion of such information there, in a graphical way.
4.-At Section 2.5, lines 237-240, it should be made clearer that the eluate "contained EDTA" to facilitate release of the potential interactor. Also, at Section 3, line 269-270, it reads "... the majority of these proteins represent soluble residents in the lumen ...". Probably grammatically is not correct to use there the term "majority" when referring to the candidate protein interactors, given that they are only three (endoplasmin, BiP and calreticulin-3).
Author Response
We would like to thank the reviewers for their time and helpful comments. Below we address each of the comments one by one.
Reviewer 2 Comments for the Author
This appears as a well made, easy to follow and understand work, that provides new and interesting insights on the protein partners of the subtilisin-like phytaspase, a protease involved in programmed cell death in plants. The works seems to be correctly made and contributes to the advancement of its research field. There are a series of points of this work on which the present reviewer considers that should be revised for its improvement and a better understanding:
1.-At the Introduction, lines 35-38, although it could be deduced from the context that SASP, the senescence-associated protease, mentioned in these lines is a subtilisin-like protease, it would be desirable to explicitly indicate such identification there, at such point.
Answer: Done. Please see p. 1 in the revised version of the manuscript.
Also, at lines 40-44 it is commented that phytaspases are distinguished by their strict cleavage specificity after an aspartate residue, after a tripeptide, motif; here, again, it would be wise to add the word "substrate" before "cleavage specificity" (or equivalent) to avoid possible doubts on the receiver of such action, and more if it is taken into account that such enzymes experiments auto-processing events.
Answer: Done. Please see p. 1 in the revised version of the manuscript.
Also, in the same paragraph, at lines 43-44, it is commented that it can be said that such proteases, phytaspases, "are processive rather than digestive" due to such type of recognition. Given that "processivity" is a term that, in such context, could have several variants and meaning, it would be advisable to better explain it there and/or in another point of the manuscript. Also, the term "makes" there is a bit confusing, and it would be better substitute it.
Answer: Done. Please see p. 1 in the revised version of the manuscript.
2.-At different points of the work there are citations and comments on subtilases/phytaspases from A. thaliana and from N. tabacum. However, the research work is actually performed on N. benthamiana leaves. Authors should explain at the Introduction (and /or perhaps also at the Discussion) why they finally selected N. benthamiana to do the work
Answer: N. tabacum phytaspase is a prototype of phytaspases. We have chosen a closely related N. benthamiana species for our experiments because N. benthamiana is a widely used model plant for many aspects of plant biology, and because the BioID approach has been optimized for this plant (see ref. 20 in our original text). We now added our explanation to the text, please see p. 9 in the revised version of the manuscript.
3.-Figure1 displays the sketch of the proteins and protein constructs used in this work for the identification of phytaspase interactors. However, the detailed structure of such constructs, grossly exemplified by the depiction of "TurboID" domain in the sketches of N-PhyTurboID and SP-turboID, in subFig 1a, does not provide enough detailed information about, requiring the need to get such information elsewhere. Given that such Figure is not crowded, this reviewer would suggest the inclusion of such information there, in a graphical way.
Answer: While constructing the recombinant proteins, we did our best to preserve their native structure and to avoid unnecessary additions. We would therefore suggest to provide information on the short (3 amino acid long) spacer between the NtPhyt and TurboID moieties in the 4.2. Plasmid constructions section, rather than to include this superfluous detail in Figure 1a. Please see p. 11 in the revised version of the manuscript.
4.-At Section 2.5, lines 237-240, it should be made clearer that the eluate "contained EDTA" to facilitate release of the potential interactor. Also, at Section 3, line 269-270, it reads "... the majority of these proteins represent soluble residents in the lumen ...". Probably grammatically is not correct to use there the term "majority" when referring to the candidate protein interactors, given that they are only three (endoplasmin, BiP and calreticulin-3).
Answer: Done. Please see p. 8, 9 in the revised version of the manuscript.
An additional alteration in the manuscript:
our database search revealed a calreticulin-3 sequence that has an even better sequence coverage with mass spectrometry-identified peptides. We therefore provide this sequence (in Figure S2) and its identificator (in Table 1) in the revised version of the manuscript.
Round 2
Reviewer 1 Report
The revised version of the manuscript by Teplova et all has been partially improved: the methods and figure legends were only partially supplemented with the missing data.
So, there are still some aspects that require correction or supplementation/expansion. The methods section with legends for figures should make it possible to repeat the experiments, but also confirm the correctness of the conclusions drawn from these experiments; therefore all relevant details must be provided.
The issues that should be addressed are listed below:
- As the protein concentration was not evaluated at any experimental step, more detailed information regarding sample amounts should be provided. Please add information in the legends of each electrophoregram (Fig. 2a, Fig. 3, Fig. 4, Fig. 5) what the volumes of the loaded samples were and what amounts of plant tissue they corresponded to.
- Fig. 2 – please, provide values of phytaspase activity in units: ΔRFU/h*mg plant tissue
- Fig.6 – In the experiment of the CRT-NtPhyt binding, the Authors stated that the resin aliquots containing 0,5 μg of His-CRT were used. How the His-CRT protein content was estimated? How equivalent amounts of lysed vector-only cells were evaluated? These steps were not mentioned in the manuscript.
In this experiment, equal protein loads in the resin aliquots are crucial. The results presented in the Fig. 6 shows that NtPhyt bound also to the control resin. It is important to demonstrate that higher amount of NtPhyt bound to His-CRT-resin compared to the control resin loaded with E. coli lysate was really due to specific binding CRT-NtPhyt, and not due to unequal amounts of protein loaded into the resin. In my opinion, protein concentration in each lysate should be determined and equal amounts should be loaded into the resin.
Alternatively, in vitro binding experiments on purified proteins would be a very good proof of specificity of CRT-NtPhyt interactions.
- Please, provide also statistical analysis of the differences in the NtPhyt activity.
Author Response
We would like to thank the reviewer for helpful comments. Below we address each of the comments one by one.
The revised version of the manuscript by Teplova et all has been partially improved: the methods and figure legends were only partially supplemented with the missing data.
So, there are still some aspects that require correction or supplementation/expansion. The methods section with legends for figures should make it possible to repeat the experiments, but also confirm the correctness of the conclusions drawn from these experiments; therefore all relevant details must be provided.
The issues that should be addressed are listed below:
- As the protein concentration was not evaluated at any experimental step, more detailed information regarding sample amounts should be provided. Please add information in the legends of each electrophoregram (Fig. 2a, Fig. 3, Fig. 4, Fig. 5) what the volumes of the loaded samples were and what amounts of plant tissue they corresponded to.
Answer: Done; please see legends to the corresponding Figures.
- 2 – please, provide values of phytaspase activity in units: ΔRFU/h*mg plant tissue
Answer: Done; please see the modified Figure 2b with the requested values.
- 6 – In the experiment of the CRT-NtPhyt binding, the Authors stated that the resin aliquots containing 0,5 μg of His-CRT were used. How the His-CRT protein content was estimated? How equivalent amounts of lysed vector-only cells were evaluated? These steps were not mentioned in the manuscript.
In this experiment, equal protein loads in the resin aliquots are crucial. The results presented in the Fig. 6 shows that NtPhyt bound also to the control resin. It is important to demonstrate that higher amount of NtPhyt bound to His-CRT-resin compared to the control resin loaded with E. coli lysate was really due to specific binding CRT-NtPhyt, and not due to unequal amounts of protein loaded into the resin. In my opinion, protein concentration in each lysate should be determined and equal amounts should be loaded into the resin.
Alternatively, in vitro binding experiments on purified proteins would be a very good proof of specificity of CRT-NtPhyt interactions.
Answer: We agree with the reviewer that equal protein load is essential to evaluate NtPhyt binding correctly. We have always kept this in mind, and to illustrate this point we now added Figures S3 a and b as a proof of our statements.
- Please, provide also statistical analysis of the differences in the NtPhyt activity.
Answer: Done; please see the modified Figure 6 and its legend, and Materials and Methods section 4.7.
Reviewer 2 Report
In this reviewer's opinion, authors fulfilled all the requests made on the previous version
Round 3
Reviewer 1 Report
The revised version of the manuscript by Teplova et all has been corrected according to the suggestions and I recommend to accept it in the present form.